# Effects of Physical Exercise on Cerebral Blood Velocity in Older Adults: A Systematic Review and Meta−Analysis

**DOI:** 10.3390/bs13100847

**Published:** 2023-10-16

**Authors:** Tiago Paiva Prudente, Henrique Nunes Pereira Oliva, Isabela Oliveira Oliva, Eleazar Mezaiko, Renato Sobral Monteiro-Junior

**Affiliations:** 1School of Medicine, Universidade Federal de Goiás, Goiânia 74690-900, GO, Brazil; tiagopaiva@discente.ufg.br; 2Department of Psychiatry, Yale University School of Medicine, New Haven, CT 06510, USA; henrique.oliva@yale.edu; 3Postgraduation Programme of Health Sciences, Universidade Estadual de Montes Claros, Montes Claros 39401-089, MG, Brazil; 4School of Medicine, Centro Universitario FIPMoc (UNIFIPMoc), Montes Claros 39408-007, MG, Brazil; isabelaoliveiraoliva@gmail.com; 5School of Dentistry, Universidade Federal de Goiás, Goiânia 74690-900, GO, Brazil; eleazarmezaiko@discente.ufg.br; 6Postgraduation Programme of Neurology/Neuroscience, Universidade Federal, Niterói 24020-141, RJ, Brazil; 7Research and Study Group in Neuroscience, Exercise, Health and Sport—GENESEs, Physical Education Department, Universidade Estadual de Montes Claros, Montes Claros 39401-089, MG, Brazil

**Keywords:** transcranial Doppler ultrasound, neuroimaging, cerebral blood velocity, cerebral hemodynamics, physical exercise, older adults, meta−analysis

## Abstract

As the older population grows, there is an increasing interest in understanding how physical exercise can counteract the changes seen with aging. The benefits of exercise to general health, and especially to the cardiovascular system, have been a topic of discussion for decades. However, there is still a need to elucidate the effects of training programs on the cerebrovascular blood velocity in older people. This systematic review and meta-analysis aimed to investigate the effect of physical exercise on the cerebral blood velocity in older people (PROSPERO CRD42019136305). A search was performed on PubMed, Web of Science, EBSCO, ScienceDirect, and Scopus from the inception of this study to October 2023, retrieving 493 results, of which 26 were included, analyzing more than 1000 participants. An overall moderate risk of bias was found for the studies using the Cochrane risk-of-bias tools for randomized and non-randomized clinical trials. The pooled results of randomized trials showed that older people who underwent physical exercise presented a statistically significant increase in cerebral blood velocity (3.58; 95%CI = 0.51, 6.65; *p* = 0.02). This result indicates that physical exercise is important to help maintain cerebral health in older adults.

## 1. Introduction

The aging brain naturally goes through striking changes, such as cortical atrophy, volume loss, and widening of sulci. The degree and rate in which these events happen are directly related to pathological processes, such as cognitive decline, infarcts, and white matter abnormalities [1]. Estimates show that such changes start occurring at age 40 and can rapidly increase after age 70 [1]. Hence, in a world where the overall life expectancy is around 73, it is crucial to understand the factors that can slow down these processes [2].

One important factor that contributes to maintaining brain health is the patency of the blood supply and the integrity of the vasculature of the neuronal tissue [3]. Changes that affect this system have been linked to the development of neurodegenerative disorders, such as Alzheimer’s disease (AD) and Parkinson’s disease [3]. Studies have shown that cerebral blood flow (CBF) in AD is globally reduced, and the degree of these changes is correlated with the clinical severity of the disease [4,5]. Consequently, preventing reductions in CBF could be an important way to avoid or delay the appearance of neurodegenerative disorders.

A well-known intervention to achieve this goal is regular exercise. Research has shown benefits from different types of training and exercise for a broad range of neurologic/mental health conditions in different populations, both in younger individuals [6,7] and older adults [8,9,10]. Such benefits are likely related to the cardiovascular changes that arise from exercising, comprising blood volume, blood vessel, and circulatory adaptations [11,12]. The association between the benefits of exercises according to sex have also been explored with varying results, suggesting there might be no difference between men and women after resistance exercises [13], while others demonstrated that women might present impaired cerebral reactivity compared to men [14,15]. 

Studies have demonstrated that regular exercising increases the expression of antioxidant enzymes and anti-inflammatory cytokines, which is related to decreased cardiovascular risk, reduced atherosclerotic processes, and regulation of blood flow to vital organs, including CBF [16]. Exercises also promote an increase in CBF, as the body attempts to maintain adequate oxygen and nutrient delivery, as well as remove metabolic end-products [17]. Such adaptations have been linked to enhanced executive function [6], and are directly related to the intensity and duration of exercises. For example, low- and moderate-intensity aerobic activities show an overall increase in CBF in various arteries, while high-intensity exercises either plateau or decrease CBF likely to maintain thermoregulation [18]. 

Various methods have been used to analyze CBF, such as magnetic resonance imaging (MRI) and transcranial Doppler ultrasound (TCD) [12]. Despite their underlying principle differences, agreement has been shown in the analysis of CBF when both were compared [19]. The advantages of TCD over MRI are its wider availability, easier accessibility, and simpler method [20,21]. Considering these advantages, numerous studies have explored cerebrovascular circulation in relation to physical exercise. Populations with varying degrees of health vulnerability or morbidity, such as older adults, have been subjects of investigation [22,23,24]. A recent systematic review has shown variable results comparing CBF before and after exercising interventions in different populations, measured both by MRI and TCD. However, only a descriptive analysis was performed in this publication [25].

Even though various studies have been carried out using TCD, the real impact of short-time or long-term exercises in the CBF outcomes in the older adults is still unclear. Contrasting and variable results have also been shown in other publications. Some have presented data confirming that CBF—also described after evaluation of cerebral blood velocity (CBV)—is increased in older adults after months of regular exercises [26,27]. On the other hand, some publications have not found this association [28,29]. Another recent systematic review also attempted to evaluate the relationship between exercises and CBF. However, only one database was searched, and no meta-analysis was performed [30]. 

Hence, the aim of this study was to perform a more comprehensive search to pool results of published papers and perform a meta-analysis to understand the effects of physical exercise on the CBV in older adults measured by TCD. We chose to include studies that analyze cerebral hemodynamics through CBV because this method is noninvasive, easier to perform, widely available, and allows real-time evaluations [19,21].

## 2. Materials and Methods

We performed a systematic literature review and meta-analysis on the effects of physical exercise on the CBV in the older population. The study was conducted and reported following the Preferred Reporting Items for Systematic Reviews and Meta-analyses (PRISMA) (Appendix A) [31]. A review protocol was developed and registered with PROSPERO (registration number CRD42019136305).

### 2.1. Study Eligibility

The inclusion criteria for this review were defined by applying the PICO framework (Population: older adults, as defined by each study, or mean age of study population equal or above 60 years; Intervention: any form of physical exercise, such as resistance/strength training, walking, running, and biking; Comparator: pre-exercise values or values from a control group that did not undergo physical exercise; Outcome: CBV either in the internal carotid arteries (ICAv) or in the middle cerebral arteries (MCAv) measured by TCD). Only experimental studies published as full articles were included. No restrictions were made in terms of date or language of publication. 

Studies were excluded for the following reasons: participants were not older adults; there was no physical exercise intervention; the study did not provide measurements of interest; the paper could not be found; the publications were not full articles (e.g., posters, abstracts) or came from observational studies. 

### 2.2. Search Strategy

A systematic search was conducted in the electronic databases PubMed, Web of Science, ScienceDirect, EBSCO, and Scopus from inception until October 2023 in order to identify relevant eligible studies for this review. The search strategy was developed with keywords based on the PICO framework guidelines. We combined terms such as “transcranial Doppler”, “cerebral blood flow”, “cerebral blood velocity”, “exercise”, and “older people”. The adapted strategy for each database can be found in the Appendix A.

### 2.3. Study Screening

The screening, inclusion, and exclusion was facilitated using Rayyan, a web app for systematic reviews [32,33]. Records from databases were imported into Rayyan, where duplicates were removed according to titles and authors. Afterward, using the “blinding” tool of the web app, studies were independently screened against the PICO criteria for inclusion or exclusion by two of the authors (TPP and IOO). Contrasting decisions were solved by a third author (HNPO) after blinding was turned off. Eligible studies were put forward for data extraction.

### 2.4. Data Extraction and Coding

The data were extracted independently by two of the authors (TPP and EM). The following data were extracted from the selected reports and explored for further analysis: (1) author, year; (2) number of participants, including those in the experimental and control groups; (3) study design; (4) mean age of participants; (5) previous fitness status of participants; (6) type of intervention; (7) intervention schedule and duration; and (8) experimental results. 

### 2.5. Risk of Bias in Individual Studies

The revised Cochrane risk-of-bias tool for randomized trials (RoB2) [34] was utilized to investigate the potential risk of bias in the randomized studies analyzed. The tool provides a range of low to high results in evaluating bias in papers. Factors that could contribute to bias include selection bias resulting from the randomization process, performance bias from deviations in the intended intervention, attrition bias from missing outcome data, detection bias from the measurement of outcome, and reporting bias from the selection of the reported result. 

For the quasi-experimental studies included, the Risk Of Bias In Non-Randomized Studies—of Interventions (ROBINS-I) [35] was used. The tool comprises seven domains: bias due to confounding, in the selection of study participants, in the classification of interventions, due to deviations from intended interventions, due to missing data, in the measurement of outcomes, and in the selection of the reported results. The categorization of the overall risk of bias followed the guidelines proposed by each of the tools.

### 2.6. Certainty Assessment

The certainty of evidence was assessed using the Grades of Recommendation, Assessment, Development, and Evaluation (GRADE). This tool evaluates five domains: risk of bias, inconsistency, indirectness, imprecision, and publication bias. The certainty for the body of evidence is categorized as high, moderate, low, or very low [36].

### 2.7. Data Synthesis

For each included study, relevant data were extracted, effect sizes were calculated based on reported means, standard deviations, and sample sizes, and if necessary, authors were contacted to provide missing data. The mean difference was used to report effect size, as all included studies presented results as cm/s. We also interpreted effect size according to the result obtained, considering it as small (d ≤ 0.2), medium (d ≈ 0.5), or large (d ≥ 0.8) [37]. For studies including more than one comparison with the same control group, we separated the “shared” group equally with smaller sample sizes to overcome the unit-of-analysis error [38]. 

Heterogeneity across studies was assessed using the I^2^ statistic and considered not important (I^2^ 0–40%), moderate (I^2^ 40–60%), substantial (I^2^ 60–90%), and considerable heterogeneity (I^2^ 90–100%). A fixed-effects model was employed for synthesizing the data due to the absence of significant heterogeneity among the included studies. Subgroup analyses were performed to explore potential sources of variation, such as variations in study design, physical activity regimens, and participant characteristics.

In order to examine the publication bias, a visual evaluation of the funnel plots for each comparison analyzed was conducted. This involved positioning each study on a graph based on its standard error (precision) and effect size [39]. The forest plots and funnel plots were generated using the Review Manager 5 computer program (RevMan^®^ 5.4) [40]. 

## 3. Results

### 3.1. Study Selection

Our search retrieved 493 results, of which 52 were duplicates. After the title and abstract screening, 411 were excluded. Full text evaluation of the remaining articles yielded 23 studies that met the inclusion criteria. Additionally, three articles were later retrieved after evaluation of the reference lists of other published studies. Hence, 26 publications were included in our systematic review (Figure 1). If necessary, we asked the authors for additional information via email to include data in our meta-analysis. If we could not obtain a response within two weeks after the first email, the publication was included only in the qualitative analysis. After that process, we were able to meta-analyze 12 studies.

### 3.2. Study Characteristics

Overall, 1159 participants were analyzed in the included studies, of which approximately 56% were females, from North America, South America, Europe, Asia, and Australia. Of the included papers, thirteen used a quasi-experimental design, while eleven were randomized controlled trials (RCT), and two applied a randomized crossover design. The mean age of participants was above 60 across all studies, with a maximum mean of 78 years. Most of the studies (76%) involved aerobic exercises, such as running and cycling. Among these, one publication investigated the differences between moderate-intensity exercises and high-intensity interval training (HIIT) [41]. Other types of exercise, such as taekwondo [29] and Baduanjin [42] training, were also used. Handgrip exercises were examined in two studies [43,44].

The exercising period varied from single-time activities to 12 months of regular activities. The schedules were also variable, ranging from 20 to 40 min of exercises 3 times a week, to 60 min 5 days/week. During the activities, the maximal percentage of heart rate aimed varied between 30% and 90%. It is important to note that not all studies included data regarding heart rate, as for some exercises, such as handgrip, this variable was not of interest (Table 1).

In sixteen of the included studies, the populations comprised previously sedentary or low-active individuals [24,27,29,41,42,45,46,47,50,52,56,57,58,59,60]. Three other publications included only previously active participants, defined as those who practiced aerobic activities 2 to 3 days/week [51], aerobic activities three or more days/week for >30 min each [53], or runners with a mean of eight weekly hours of training [26]. The remaining studies did not mention the previous activity status of participants [28,43,44,48,49,54,55].

Additionally, all included studies analyzed the MCAv (either bilaterally or unilaterally). Other variables, such as the pulsatile index, the end diastolic blood flow velocity, and the peak systolic blood flow velocity were also mentioned in some publications [24,27,29,43,45,51,57]. Two studies [27,53] also analyzed the ICAv. Additionally, results were presented mainly by showing the mean MCAv or ICAv in centimeters/second (cm/s). However, some of them presented data differently, such as median and interquartile range [52], median or mean and 95%CI [27,47], and percentage changes [48].

### 3.3. Risk of Bias within Studies

The risk of bias can be found in Figure 2 and Figure 3. For the non-randomized controlled trials (quasi-experimental), we judged that there was an overall moderate risk of bias. This result was mainly related to bias due to the selection of participants into the study, as eight studies did not describe where the recruited population came from or how it was selected [23,26,43,44,45,51,53,55]. Additionally, none of the quasi-experimental studies described assessor blinding, consequently receiving a moderate risk of bias for measuring the outcome [24,26,28,43,44,45,46,48,51,53,54,55].

As for the randomized controlled trials and randomized crossover trials, most of the studies were also judged to have an overall moderate risk of bias. Half of them did not describe how the randomization process was carried out, consequently being judged as having “some concerns” [29,47,49,50,52,59]. One study was also deemed to have some concerns for the missing outcome data domain, as it did not clearly explain why two participants did not complete the study [52]. Lastly, seven publications did not report assessor blinding [29,41,47,49,50,52,59].

### 3.4. Results of Individual Studies

Among the included studies, sixteen showed significant increases in MCAv after exercise in their analyses [24,26,27,41,42,43,44,45,48,49,51,54,57,58,59,60]. Nine of them used more than eight weeks of regular physical activity [24,27,41,42,45,49,57,58,59], while the others used single bouts of exercises for a maximum of four times [26,43,44,48,51,54,60].

As for the studies that yielded no significant results, seven applied more prolonged exercise regimens [23,28,29,47,52,55,56], whereas the others also explored single-time activities [50,53]. It is important to note that one of the publications mentioned that exercise increased MCAv but did not state whether the results were significant [46]. 

Additionally, two of the included studies submitted participants only to single-time handgrip exercises [43,44], both showing significant increases from baseline. Two other studies applied varied schedules of prolonged sitting, sometimes interrupted by watching television or by cognitive stimulation [50,60]. Between these, results were different, showing significant differences in one study [60] and no difference in the other [50].

### 3.5. Certainty Assessment

The certainty of the evidence for the RCTs was considered high, as the risk of bias, inconsistency, and indirectness were considered not serious. The imprecision domain was considered serious due to the large confidence interval generated in the forest plot (Appendix A).

### 3.6. Synthesis of Results

Overall, our meta-analysis showed that exercise significantly increases MCAv in the older population when evaluated with TCD (1.90; 95%CI = 0.89, 2.91; *p* < 0.01), with a large effect size (Figure 4). A separate analysis for the quasi-experimental pre–post trials, in which participants were sedentary and engaged in aerobic exercises during the study period, also yielded significant results (2.16; 95%CI = 0.45, 3.86; *p* = 0.01). On the other hand, meta-analysis for the randomized crossover trials did not yield significant changes (1.20; 95%CI = −0.22, 2.61; *p* = 0.10). The RCTs, which involved prolonged aerobic exercise for previously sedentary older adults, also showed significant increase in MCAv (3.58; 95%CI = 0.51, 6.65; *p* = 0.02). Lastly, the pooled data for the two quasi-experimental studies in which all participants were previously active and performed single-time aerobic exercises before analysis did not show significant results (4.30; 95%CI = −1.08, 9.68; *p* = 0.12). Heterogeneity was considered “not important” across all comparisons (I^2^ < 40%). Some studies could not be included in the meta-analysis because they did not report the previous physical preparation of the participants, their interventions were not comparable to others, or we were unable to obtain the mean and standard deviation of the individual results.

Funnel plot assessing publication bias can be seen in Appendix A.

**Figure 4 behavsci-13-00847-f004:**
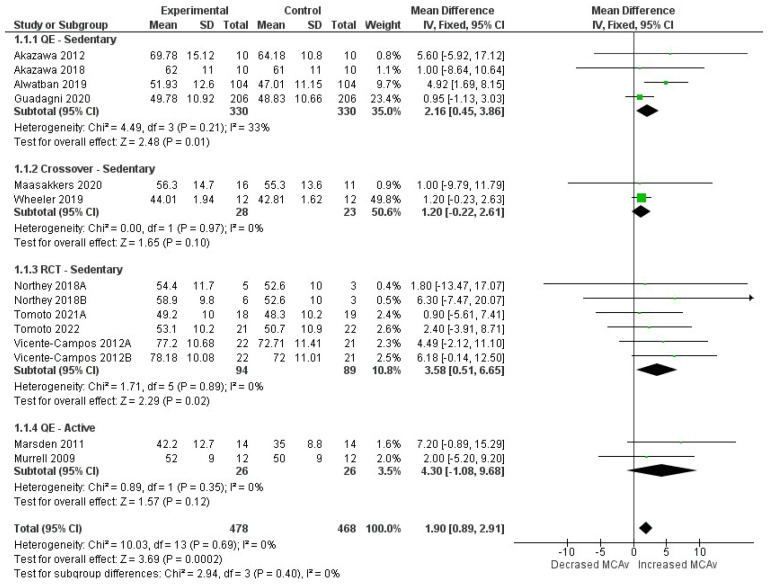
Forest-plot pooling data of all included studies [24,26,41,45,46,50,51,56,58,59,60]. From top to bottom: (1.1.1) Quasi-experimental studies in which all participants were previously sedentary and performed aerobic exercises during the study period; (1.1.2) Randomized crossover trials in which all participants were previously sedentary and were submitted to mixed situations of prolonged sitting interrupted (or not) by walking; (1.1.3) Randomized controlled trials in which all participants were previously sedentary and performed aerobic exercises during the study period; (1.1.4) Quasi-experimental studies in which all participants were previously active and performed single-time aerobic exercise during the study. Figure legend: 95%CI, 95% confidence interval; df, degrees of freedom; IV, inverse variance; QE, quasi-experimental studies; RCT, randomized controlled trial; SD, standard deviation of the middle cerebral artery velocities. Northey 2018A [41], moderate-intensity continuous training; Northey 2018B [41], high-intensity interval training; Tomoto 2021A [56] was used to differentiate from Tomoto 2021B [57], which was not meta-analyzed; Vicente-Campos 2021A [59], right middle cerebral artery; Vicente-Campos 2021B [59], left middle cerebral artery.

## 4. Discussion

This systematic review and meta-analysis gathered results of published randomized and non-randomized clinical trials that aimed to investigate the effects of physical exercise on the CBV in older adults using TCD. Overall, we found that physical exercise significantly increased MCAv.

The MCAs supply 80% of the blood flow to the brain [61], which makes the measurement of MCAv an interesting and indirect method of analyzing the whole CBV. Data shows that MCAv indeed correlates well with the corresponding CBV in the areas perfused by these arteries [62]. Hence, the measurement of this parameter using TCD is a widely used method, which estimates MCAv through the temporal bone, where thinner layers allow for better accuracy [63].

One important aspect regarding the relationship between physical exercise and CBV is the type of activity under investigation. For instance, our meta−analysis showed that regular aerobic exercises significantly increased MCAv in previously sedentary older individuals, while single−time aerobic exercises and short−term walking did not yield significant results. This is likely because regular aerobic exercises involves rhythmic movement of large muscle groups, and its sustained nature allows for gradual adaptation of the cardiovascular system, leading to improved plasma volume, stroke volume, and a consequent more efficient oxygen−carrying capacity of the blood [11]. Additionally, chronic exercise is known to promote angiogenesis through the formation of new capillaries from pre−existing blood vessels [64,65]. Thus, active older individuals might present better measurements of CBV compared to sedentary people due to baseline adaptations that they have already experienced.

On the other hand, short−term exercises generate different responses and may not elicit significant changes in CBV. During these bouts of exercises, up to 80% of the cardiac output is directed to skeletal muscles and to the skin, which also extract a greater amount of oxygen from the circulating blood [11]. Hence, the diverse mechanisms of adaptation during acute and chronic exercise stimulate varying levels of cardiovascular adaptation. Furthermore, the concept of the “dose–response” relationship between exercise and CBV warrants further investigation. The specific threshold at which the benefits saturate, or plateau, remains an area of ongoing research. 

Additionally, exercise intensity is also an important aspect. The regular aerobic exercises in the included studies ranged from 75 to 360 min/week, some of them with increasing frequency during the study period. In addition, most of the studies aimed at reaching at least 70% of the maximal heart rate. Such variation might influence the magnitude of the observed effects on CBV. For example, exercises that demand roughly 60% of the maximal oxygen uptake have been shown to increase CBV, while higher demands decrease this parameter because of hyperventilation−induced cerebral vasoconstriction [12]. Additionally, in one of the included studies, HIIT showed increased MCAv compared to control, while moderate−intensity exercising did not [41], which corroborates the importance of exercise intensity. Hence, the mentioned physiological changes may translate into more efficient oxygen and nutrient delivery to brain tissues, fostering neuroplasticity and cognitive performance.

However, it is important to note that the relationship between exercise intensity and CBV is likely to be nonlinear and subject to individual variations. Factors such as underlying health conditions can also impact cerebral hemodynamics. Congestive heart failure has been shown to impair cerebral autoregulation due to the limited compensatory mechanisms of arteriolar vasodilation, which ultimately decreases CBV [66]. Studies have also found a interesting association between CBV and heart failure severity [67,68]. In addition, a systematic review has shown that HIIT presents variable effects on cerebrovascular function, including MCAv [69]. This modality of exercise has gained attention due the shorter duration of practice required to achieve health benefits [70]. Thus, more studies are necessary to investigate the long-term implications of HIIT on cerebrovascular function and to better understand the specific mechanisms underlying the observed variability in its effects on MCAv. 

The study design may also interfere with results. For example, studies that perform intraindividual comparisons pre−post physical exercises are more likely to better adjust confounders and individual variability by turning each person into their own control. On the other hand, studies that carry out interindividual comparisons might produce more generalizable results and consider higher diversity of characteristics [71]. Thus, pooling results of each of these separately is an important strategy to account for the possible divergencies. In this sense, our meta−analysis showed a significant increase in MCAv for both situations, represented by the RCTs (interindividual comparisons) and quasi−experimental trials (intraindividual comparisons). 

The present work offers pooled results on CBV assessed with TCD. However, information on CBF could provide further evidence. CBF is defined as the volume of blood that passes through a specific quantity of brain tissue during a particular period of time. Classically, CBF is measured in units of mL of blood per 100 g of tissue per minute. Computed tomography and MRI perfusion techniques measure the quantity of blood that flows through a particular volume of brain tissue, i.e., a voxel, rather than a particular mass [72]. 

Neurophysiological investigations have supplied proof that even a single bout of physical exercise heightens brain function in distinct brain regions, like the hippocampal and frontal areas. Both clinical and preclinical analyses have employed diverse brain imaging and electrophysiological methodologies, including electroencephalography, functional MRI, and transcranial magnetic stimulation [73]. These alterations are linked with increased blood circulation and simultaneous cognitive enhancements in individuals undertaking immediate exercise, irrespective of their age [73,74]. Moreover, there is an indication that a single exercise session could foster both restraint and activation in the motor cortex, correlating with progress in motor learning and possible amplifications in cortical adaptability [73], which could be important interventions for fall prevention and for modulating some degenerative diseases, such as Parkinson’s disease, in older adults [75,76]. The outcomes from these alternative techniques serve as valuable tools and complement the current findings obtained from TCD.

Lastly, it is important to acknowledge some limitations of the present study. In order to maintain a focus on the older population in a consistent manner, only studies with older adults, as defined by each study, or with an age of study population equal or above 60, were included. While this approach allowed for a more standardized analysis, it is possible that additional age ranges could have provided further insights. As a result, several studies that investigated exercise intervention in middle-aged adults were excluded from this review. Additionally, due to the heterogeneity among study designs and physical preparation of participants pre-exercise, subgroup analysis had to be performed, in addition to the overall meta-analysis. Thus, the number of studies per subgroup was considerably small and inclusion of some publications to the meta-analysis was not possible. Moreover, the blinding of participants was impracticable given the nature of intervention considered in the trials. Previous work has rated such a bias domain as having low risk, due to the impossibility of blinding participants and evaluators [77]. Also, the present meta-analysis pools studies investigating CBV by TCD, but not CBF, or outcomes from other more sophisticated imaging techniques such as MRI, potentially capable of providing further assessment of vasculature. Lastly, as none of the retrieved results that studied the impact of resistance/strength exercises met the inclusion criteria, we were unable to analyze the effects of this type of activity.

## 5. Conclusions

The present meta-analysis has revealed the significance of exercise in preserving cerebral health among older adults, notably by enhancing CBV. However, the implications of these findings extend beyond the older adult population. Transcranial Doppler, as employed in these studies, presents a promising avenue for assessing both brain and cardiovascular health in a broader context, encompassing individuals of varying age groups. While our analysis primarily focused on older adults, future research could delve deeper into the exploration of neurobiological mechanisms that may mechanistically underpin the influence of physical exercise on cerebral blood flow. This avenue of investigation not only holds promise for elucidating the intricate relationship between exercise and cerebral health but also for its potential application as a diagnostic and preventive tool in broader populations. Understanding how exercise impacts cerebral blood flow can have far-reaching implications for health promotion and disease prevention across the lifespan. In summary, our findings underscore the importance of exercise in maintaining cerebral health among older adults, opening up new possibilities for utilizing TCD and its results as valuable metrics in assessing brain and cardiovascular health across diverse age groups.

## Figures and Tables

**Figure 1 behavsci-13-00847-f001:**
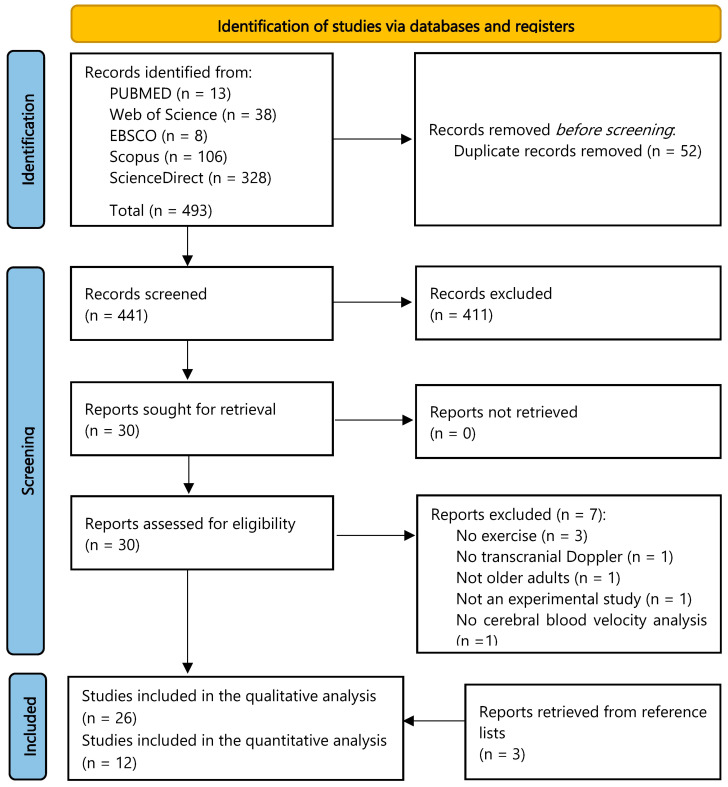
PRISMA flow diagram of the study selection process.

**Figure 2 behavsci-13-00847-f002:**
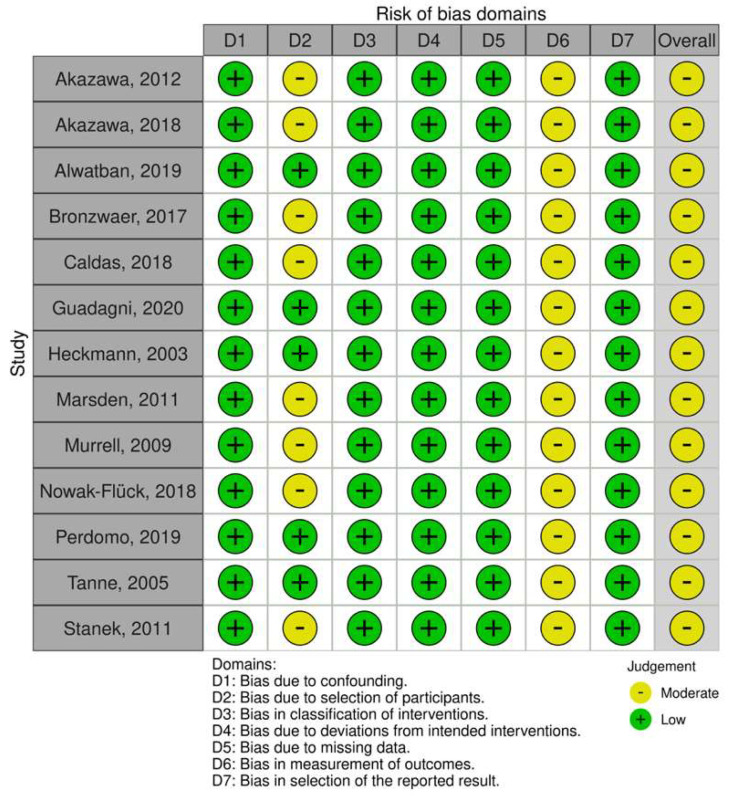
Risk of bias of the non-randomized clinical trials using the ROBINS-I tool [24,26,28,43,44,45,46,48,51,53,54,55].

**Figure 3 behavsci-13-00847-f003:**
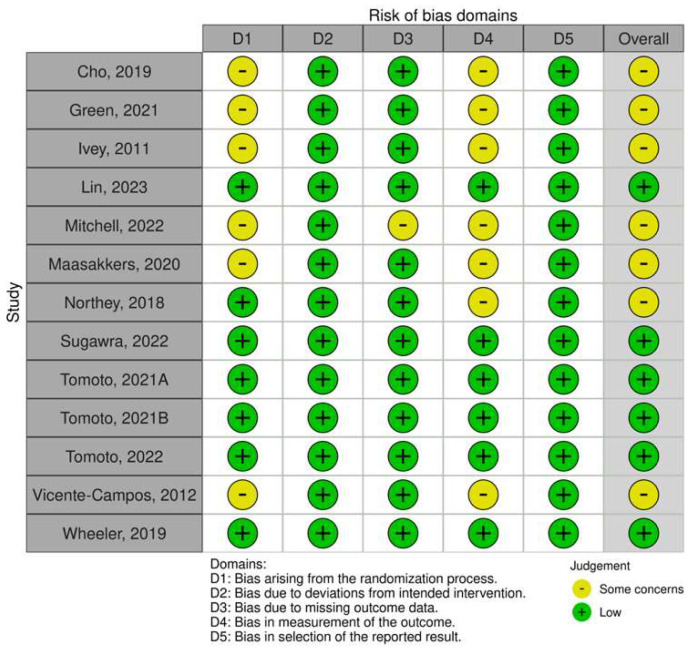
Risk of bias of the randomized studies (randomized controlled clinical trials and randomized crossover trials) using the RoB2 tool [27,29,41,42,47,49,50,52,56,57,58,59,60].

**Table 1 behavsci-13-00847-t001:** Characteristics of the studies included in this systematic review.

Author, Year	Design	N	Age of Participants (Mean, SD)	Previous Status	Exercise Type	Minutes/Week	Exercise Duration (Total)	Cerebral Blood Velocity Finding
Akazawa, 2012 [45]	QE	20	Control: 61 ± 2 Exp: 60 ± 2	Sedentary	Aerobic exercise	90–360	8 weeks	↑
Akazawa, 2018 [23]	QE	10	62 ± 4	Sedentary	Aerobic exercise	140–270	12 weeks	—
Alwatban, 2019 [46]	QE	104	70.39 ± 4.78	Sedentary	Aerobic exercise	Single time	8 min	—
Bronzwaer, 2017 [43]	QE	11	72 ± 3	NR	LBNP or dynamic HG	Single time	15 min	↑
Caldas, 2018 [44]	QE	63	Control: 62.8 ± 8.6 Exp: 62.9 ± 8.7	NR	Isometric HG	Single time	3 min	↑
Cho, 2019 [29]	RCT	37	Control: 69.00 ± 4.41 Exp: 68.89 ± 4.16	Sedentary	Taekwondo	300	16 weeks	—
Green, 2021 [47]	RCT	63	Control: 60.9 ± 6.1 Land: 63.9 ± 7.7 Water: 61.8 ± 5.8	Sedentary/Low-active	Aerobic exercise (land and water)	45–180	24 weeks	Land: — Water: —
Guadagni, 2020 [24]	QE	206	65.9 ± 6.4	Low-active	Aerobic exercise	60–120	6 months	↑
Heckmann, 2003 [48]	QE	18	66.5 ± 5.8	NR	Aerobic exercise	Single time	3 min	↑
Ivey, 2011 [49]	RCT	38	Control: 61 ± 8 Exp: 62 ± 10	NR	Aerobic exercise	120	6 months	↑
Lin, 2023 [42]	RCT	102	Control: 65.35 ± 5.15 Exp: 67.68 ± 5.19	Sedentary	Baduanjin training	180	24 weeks	↑
Maasakkers, 2020 [50]	RCOT	22	78 ± 5.3	Sedentary	4 conditions	1 condition/week	4 weeks	—
Marsden, 2011 [51]	QE	14	71 ± 10	Active	Aerobic exercise	Single time	Variable	↑
Mitchell, 2022 [52]	RCT	20	64.95 (7.77) (Median, IQR)	Sedentary/Low-active	Aerobic exercise	150	26 weeks	
Murrell, 2009 [26]	QE	12	65 ± 5	Active	Aerobic exercise	Single time	4 h	↑
Northey, 2018 [41]	RCT	17	Control: 61.5 ± 7.8 MOD: 60.3 ± 8.1 HIIT: 67.8 ± 7.0	Sedentary/Low-active	Aerobic exercise (MOD and HIIT)	60–90	12 weeks	MOD: — HIIT: ↑
Nowak-Flück, 2018 [53]	QE	9	65.5 ± 2.8	Active	Aerobic exercise	Single time	Variable	
Perdomo, 2019 [54]	QE	72	70.1 ± 4.7	NR	Aerobic exercise	Single time	~10 min	↑
Stanek, 2011 [55]	QE	42	68.17 ± 9.03	NR	Aerobic exercise	90	12 weeks	—
Sugawara, 2022 [27]	RCT	73	Control: 68 ± 5 Exp: 69 ± 6	Sedentary	Aerobic exercise	75–200	12 months	↑
Tanne, 2005 [28]	QE	23	63 ± 13	NR	Aerobic exercise	100	18 weeks	—
Tomoto, 2021A [56]	RCT	37	Control: 64.8 ± 6.6 Exp: 64.6 ± 5.9	Sedentary	Aerobic exercise	75–200	12 months	—
Tomoto, 2021B [57]	RCT	48	Control: 66.1 ± 6.8 Exp: 64.8 ± 6.4	Sedentary	Aerobic exercise	75–200	12 months	↑
Tomoto, 2022 [58]	RCT	43	Control: 67.8 ± 4.9 Exp: 68.2 ± 5.3	Sedentary	Aerobic exercise	75–200	12 months	↑
Vicente-Campos, 2012 [59]	RCT	43	Control: 64 ± 5 Exp: 64 ± 4	Sedentary	Aerobic exercise	150–200	7 months	↑
Wheeler, 2019 [60]	RCOT	12	70 ± 7	Sedentary/Low-active	4 conditions	1 condition/week	4 weeks	↑

↑, increased cerebral blood velocity in the experimental group; —, no significant difference in cerebral blood velocity; ~, approximately; CG, control group; EG, experimental group; HG, handgrip; HIIT, high-intensity interval training; IQR, interquartile range; LBNP, lower body negative pressure; min, minutes; MOD, moderate-intensity continuous training; N, sample size; NR, not reported; QE, quasi-experimental; RCT, randomized controlled trial; RCOT, randomized crossover trial; SD, standard deviation.

## Data Availability

Data will be made available upon reasonable request.

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
