# Peer review of "Effects of Physical Exercise on Cerebral Blood Velocity in Older Adults: A Systematic Review and Meta−Analysis"

_behavsci, 2023, doi:10.3390/bs13100847_

Round 1

Reviewer 1 Report

An interesting review of the literature related to the effects of exercise on blood circulation in the brain in elderly people. The review is comprehensive but there are still some issues to be resolved before it can be published

Introduction:

The mechanisms by which exercise has an effect on brain blood velocity are not explained.  The authors should explain how physical exercise has antioxidant and anti-inflammatory effects, and how these mechanisms influence atherosclerosis and CBV.

Daniela, M.; Catalina, L.; Ilie, O.; Paula, M.; Daniel-Andrei, I.; Ioana, B. Effects of Exercise Training on the Autonomic Nervous System with a Focus on Anti-Inflammatory and Antioxidants Effects. Antioxidants 2022, 11, 350. https://doi.org/10.3390/antiox11020350

Materials and Methods

Lines 104-107 rephrase the sentence

Lines 111-115 Explain fully how Rayyan was used and use several bibliographical references

The results are well structured. Are figures 1 and 2 (optional) necessary?

Discussions and limitations are clear and well-founded

Conclusions should be expanded

Author Response

An interesting review of the literature related to the effects of exercise on blood circulation in the brain in elderly people. The review is comprehensive but there are still some issues to be resolved before it can be published.

Dear reviewer, we would like to thank you for taking the time to read our manuscript and for implementing important comments that certainly elevate the level of our paper. We tried to address your valuable suggestions to the best of our capacities and hope they improved the work.

Introduction:

The mechanisms by which exercise has an effect on brain blood velocity are not explained.  The authors should explain how physical exercise has antioxidant and anti-inflammatory effects, and how these mechanisms influence atherosclerosis and CBV.

Daniela, M.; Catalina, L.; Ilie, O.; Paula, M.; Daniel-Andrei, I.; Ioana, B. Effects of Exercise Training on the Autonomic Nervous System with a Focus on Anti-Inflammatory and Antioxidants Effects. Antioxidants 2022, 11, 350. https://doi.org/10.3390/antiox11020350

That is very interesting. We really appreciate your comment. We have added that important piece of information in lines 50-52: “Studies have demonstrated that regular exercising increases the expression of antioxidant enzymes and anti-inflammatory cytokines, which is related to decreased cardiovascular risk, reduced atherosclerotic processes, and regulation of blood flow to vital organs, including CBF [10].”

Materials and Methods

Lines 104-107 rephrase the sentence.

Thank you very much for bringing attention to this. The line is rephrased as: We combined terms such as “transcranial doppler”, “cerebral blood flow”, “cerebral blood velocity”, “exercise”, and “older people”.

Lines 111-115 Explain fully how Rayyan was used and use several bibliographical references.

            We really appreciate the opportunity to clarify it. We added an additional reference (31 - Elmagarmid, A.; Fedorowicz, Z.; Hammady, H.; Ilyas, I.; Khabsa, M.; Ouzzani, M. Rayyan: A Systematic Reviews Web App for Exploring and Filtering Searches for Eligible Studies for Cochrane Reviews. In Proceedings of the Evidence-Informed Publich Health: Opportunities and Challenges; Abstracts of the 22nd Cochrane Colloquium: Hyderabad, India, 2014) and also added a better explanation of how the tool was used: “Records from databases were imported into Rayyan, where duplicates were removed according to titles and authors. Afterward, using the “blinding” tool of the web app, studies were independently screened against the PICO criteria for inclusion or exclusion by two authors (TPP and IOO). Contrasting decisions were solved by a third author (HNPO) after blinding was turned off.

The results are well structured. Are figures 1 and 2 (optional) necessary?

Thank you very much! As the study selection process is an integral part of the systematic review, as well as the risk of bias, and as they are part of the PRISMA guidelines and checklist, we believed it would be important to display them in the manuscript.

Discussions and limitations are clear and well-founded.

We really appreciate your comment!

Conclusions should be expanded.

Thank you very much. We expanded the conclusion, which now is the following: “The present meta-analysis has revealed the significance of exercise in preserving cerebral health among older adults, notably by enhancing CBV. However, the implications of these findings extend beyond the older adult population. Transcranial doppler, as employed in these studies, presents a promising avenue for assessing both brain and cardiovascular health in a broader context, encompassing individuals of varying age groups. While our analysis primarily focused on older adults, future research could delve deeper into the exploration of neurobiological mechanisms that may mechanistically underpin the influence of physical exercise on cerebral blood flow. This avenue of investigation not only holds promise for elucidating the intricate relationship between exercise and cerebral health but also for its potential application as a diagnostic and preventive tool in broader populations. Understanding how exercise impacts cerebral blood flow can have far-reaching implications for health promotion and disease prevention across the lifespan. In summary, our findings underscore the importance of exercise in maintaining cerebral health among older adults, opening up new possibilities for utilizing TCD and its results as valuable metrics in assessing brain and cardiovascular health across diverse age groups."

Reviewer 2 Report

Thank you for the opportunity to review this meta-analysis of the effects exercise has on CBV. The authors have described the process of conducting the bias risk and the analysis very well.  I have some comments that may help add clarity to the study.

-typo noted p. 2 line 60 '... over MRI are its wider available,..." I believe you mean "availability"

- Study eligibility- you indicate that the population for inclusion was an age of equal to or greater than 60 yr. However, when I reviewed Table 1. I noted that the mean age for a number of studies was 60y but with a standard deviation suggesting that some subjects were not equal to or greater than 60. (Akazawa 2018, Ivey 2011, Northey 2018). I suggest you rephrase inclusion criteria such that the mean age of the study samples was 60 yrs or older, so as not to suggest all subjects were 60 or older.

- 2.2 Search strategy- Curious why CBV was not one of the search keys used, since it was the dependent variable of interest. Are you sure that cerebral blood flow or cerebral circulation adequately identified all studies?

- 3.2 term identifier- I see RCT identified in this section as "randomized controlled trials. However on p. 3 (line 288 the term "randomized crossover trials is used. I would suggest that if you meant the same studies to adjust to a single indicator for RCT

- Table 1- Suggestion Last column CBV finding. In two rows you indicate results for land,water and Mod, HIIT. Could you also identify the exercisees in the Exercise type column? This confused me initially because I did not see any reference to the different exercises under the exercise type column because it only stated "aerobic exercise"

- Question about CBV- were you able to determine if any studies had subjects with abnormally low pre-training CBV? This may be an influencing factor when looking at changes with training. Normal pre training CBV values may not show significant improvements, while low initial values may see more of a change. If you have information on starting values, that may add to the clarity of the results.

3.3 Risk of bias- Could you clarify in Figures 1 & 2, perhaps in the figure description, which represented the RCT and which the quasi-experimental? I believe Fig 1 was for the Quasi and figure 2 was RCT? is that correct. it was not clear.

Figure 3. I see mean and SD data reported on the table, but I cannot find a description of what these data are. Are they CBV values? MCAv values?. Perhaps the figure description could help the reader see.

Author Response

Thank you for the opportunity to review this meta-analysis of the effects exercise has on CBV. The authors have described the process of conducting the bias risk and the analysis very well.  I have some comments that may help add clarity to the study.

Dear reviewer, we would like to thank you for taking the time to read our manuscript and for implementing important comments that certainly elevate the level of our paper. We tried to address your valuable suggestions to the best of our capacities and hope they improved the work.

- typo noted p. 2 line 60 '... over MRI are its wider available,..." I believe you mean "availability".

Thank you very much for noticing the typo. We have fixed it and now “availability” is written.

- Study eligibility- you indicate that the population for inclusion was an age of equal to or greater than 60 yr. However, when I reviewed Table 1. I noted that the mean age for a number of studies was 60y but with a standard deviation suggesting that some subjects were not equal to or greater than 60. (Akazawa 2018, Ivey 2011, Northey 2018). I suggest you rephrase inclusion criteria such that the mean age of the study samples was 60 yrs or older, so as not to suggest all subjects were 60 or older.

We really appreciate you bringing attention to this. We agree with your observation and now we write: “Population: older adults, as defined by each study, or mean age of study population equal or above 60 years”.

- 2.2 Search strategy- Curious why CBV was not one of the search keys used, since it was the dependent variable of interest. Are you sure that cerebral blood flow or cerebral circulation adequately identified all studies?

            Great point. Thank you very much for mentioning it. We reran the search so that it is more updated and included the term “cerebral blood velocity” to avoid any problem that could arise from not using it. The search retrieved few additional papers, but none that met inclusion criteria. Still, we really appreciate the concern, and we agree that the update was important. The numbers are updated and highlighted in the text and Figure 1, as well as is the search strategy in Table S2.

- 3.2 term identifier- I see RCT identified in this section as "randomized controlled trials”. However on p. 3 (line 288 the term "randomized crossover trials is used. I would suggest that if you meant the same studies to adjust to a single indicator for RCT.

Thank you for your observation. As we included only experimental studies in our paper, we had three different types of study designs included: randomized controlled trials (RCT), randomized crossover trials, and quasi-experimental trials. Because of that, the RCTs indeed are not the same, but are different from the randomized crossover trials.

- Table 1- Suggestion Last column CBV finding. In two rows you indicate results for land, water and Mod, HIIT. Could you also identify the exercisees in the Exercise type column? This confused me initially because I did not see any reference to the different exercises under the exercise type column because it only stated "aerobic exercise"

Excellent observation. We really appreciate it and agree with it. To make the table clearer, we added the type of exercise to the Exercise type column, as suggested.

- Question about CBV- were you able to determine if any studies had subjects with abnormally low pre-training CBV? This may be an influencing factor when looking at changes with training. Normal pre training CBV values may not show significant improvements, while low initial values may see more of a change. If you have information on starting values, that may add to the clarity of the results.

            We really appreciate this observation. Indeed, this would be a very important aspect that could clarify results. However, only one study (Marsden, 2011) had abnormally lowlevels, presenting a MCAv mean of 35 cm/s, while the normal mean is around 55 cm/s. Because of that, we believe this might not have been a factor influencing results.

3.3 Risk of bias- Could you clarify in Figures 1 & 2, perhaps in the figure description, which represented the RCT and which the quasi-experimental? I believe Fig 1 was for the Quasi and figure 2 was RCT? is that correct. it was not clear.

Thank you for requiring clarification on this matter. You are totally correct. We have added the legends to the figures regarding risk of bias. Figure 2 (previously identified as Figure 1 because of our confusion) has now the legend: “Figure 2. Risk of bias of the non-randomized clinical trials using the ROBINS-I tool.” Additionally, Figure 3 (previously identified as Figure 2), has the legend: “Figure 3. Risk of bias of the randomized studies (randomized controlled clinical trials and randomized crossover trials) using the RoB2 tool.”

Figure 3. I see mean and SD data reported on the table, but I cannot find a description of what these data are. Are they CBV values? MCAv values?. Perhaps the figure description could help the reader see.

Great point. Thank you for noticing that. We have added the following statement on the Figure 4 (previously Figure 3) legend: “SD, standard deviation of the middle cerebral artery velocities.

Reviewer 3 Report

Dear Authors, congrats for the manuscript. See below, some points to improve it.

In Introduction, please add more information about recent studies with different type of exercise in older adults and the relashionship with vascular performance.

If is possible, clarify the research in both genders/sex.

Also if is possible, add some information related to the physiological process under physical exercise and the vascular adaptations in older people according to the type of exercise.

Discussion

This point is the most sensitive point of the manuscript. According to the revised studies, what you can add to research in this field? In older people, wich exercise is the most powerful strategy to improve Cerebral Blood Velocity? Why we want it? Why you focus on it? BEcause are related to some cardiovascular diseases?

Because the authors talk about motor learn, but in older people, it is related to fragility? or falls?

Try to add more information.

Author Response

Dear Authors, congrats for the manuscript. See below, some points to improve it.

Dear reviewer, we would like to thank you for taking the time to read our manuscript and for implementing important comments that certainly elevate the level of our paper. We tried to address your valuable suggestions to the best of our capacities and hope they improved the work.

In Introduction, please add more information about recent studies with different type of exercise in older adults and the relashionship with vascular performance.

            Thank you so much for the comment. We have added some references to this topic: “Research has shown benefits from different types of training and exercise for a broad range of neurologic/mental health conditions in different populations, both in younger individuals [6,7] and older adults [8–10].” We delve deeper into this issue in the Discussion.

If is possible, clarify the research in both genders/sex.

            We really appreciate your suggestion. We have added a sentence that addresses the possible differences in both genders: “The association between the benefits of exercises according to sex have also been explored with varying results, suggesting there might be no difference between men and women after resistance exercises [10], while others demonstrated that women might present impaired cerebral reactivity compared to men [11,12].”

Also if is possible, add some information related to the physiological process under physical exercise and the vascular adaptations in older people according to the type of exercise.

            Thank you very much for the suggestion. We have added a sentence touching on this topics in the Introduction: “Such benefits are likely related to the cardiovascular changes that arise from exercising, comprising blood volume, blood vessel, and circulatory adaptations [11,12].” The greater details of this important topic are in the Discussion.

Discussion

This point is the most sensitive point of the manuscript. According to the revised studies, what you can add to research in this field? In older people, wich exercise is the most powerful strategy to improve Cerebral Blood Velocity? Why we want it? Why you focus on it? BEcause are related to some cardiovascular diseases? 

            Thank you very much. We totally agree that this is a very important aspect of our manuscript. In an attempt to address the essential topics you mentioned, we wrote the Discussion in a way that touches on all such topics. Although we were not able to specifically answer which type of exercise would be the most powerful for older adults (due to the varying study designs, population characteristics, and absence of some commonly practiced exercise types), we attempted to talk about the possible benefits of each one. Hence, we mention that “our meta-analysis showed that regular aerobic exercises significantly increased MCAv in previously sedentary older individuals, while single-time aerobic exercises and short-term walking did not yield significant results”, and that “short-term exercises generate different responses and may not elicit significant changes in CBV”, adding that “exercises that demand roughly 60% of maximal oxygen uptake have been shown to increase CBV, while higher demands decrease this parameter because of hyperventilation-induced cerebral vasoconstriction”.

Also, for explaining the importance of exercises, we try to discuss their cardiovascular benefits, stating that “regular aerobic exercises […] allow for gradual adaptation of the cardiovascular system, leading to improved plasma volume, stroke volume, and a consequent more efficient oxygen-carrying capacity of the blood”, adding that “chronic exercise is known to promote angiogenesis through the formation of new capillaries from pre-existing blood vessels”. We end up concluding that our work “has revealed the significance of exercise in preserving cerebral health among older adults, notably by enhancing CBV”. We hope that our Discussion and Conclusion were able to make our point clear and again we deeply appreciate your important concern.

Because the authors talk about motor learn, but in older people, it is related to fragility? or falls?

Great point. Thank you very much for allowing us to clarify this point. We have complemented the sentence, which is now read as: “Moreover, there is indication that a solitary exercise session could foster both restraint and activation in the motor cortex, correlating with progress in motor learning and possible amplifications in cortical adaptability [69], which could be important interventions for fall prevention and for modulating some degenerative diseases in older adults, such as Parkinson disease [71,72].”